# Anyons in a tight wave-guide and the Tonks-Girardeau gas

Nicolas Rougerie[1][*] and Qiyun Yang[2][†]

**1** Ecole Normale Supérieure de Lyon & CNRS, UMPA (UMR 5669)
**2** Ecole Normale Supérieure de Lyon, UMPA (UMR 5669)

[*] nicolas.rougerie@ens-lyon.fr , [†] qiyun.yang@ens-lyon.fr

## Abstract

We consider a many-body system of 2D anyons, free quantum particles with general statistics parameter $\alpha \in ]0, 2[$. In the magnetic gauge picture they are described as bosons attached to Aharonov-Bohm fluxes of intensity $2\pi\alpha$, generating long-range magnetic forces. A dimensional reduction to 1D is obtained by imposing a strongly anisotropic trapping potential. This freezes the motion in the direction of strong trapping, leading to 1D physics along the weak direction. The latter is governed to leading order by the Tonks-Girardeau model of impenetrable bosons, independently of $\alpha$.

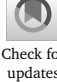

## Contents

Quasi-particle excitations of many-body systems confined to reduced dimensionalities are not in principle constrained by the symmetry dichotomy which sorts all fundamental particles into bosons and fermions [1–3, 5]. In 2D, many-body quantum wave-functions may be classified by the phase picked upon exchanging/braiding two particles. It is of the form $e^{i\pi\alpha}$ for $\alpha \in [0, 2[$, the standard cases of bosons and fermions being recovered for $\alpha = 0, 1$ respectively. Equivalently one may think of these so-called anyons in terms of standard bosons (or fermions), coupled to infinitely thin magnetic flux tubes of strength $2\pi\alpha$ (or $2\pi(\alpha-1)$). This point of view is referred to as the magnetic gauge picture [4, 6–11, 64]. In 1D there does not seem to be a unique agreed-upon model for anyonic exchange statistics. Depending on how one proceeds to quantization, they have historically been described [1, 12–14, 64] as ordinary

particles with either contact interactions (Lieb-Liniger model) or inverse-square interactions (Calogero-Sutherland model) but other formalisms exist [15–18]. In particular, the chiral BF/Kundu model [19–22] and the anyon-Hubbard model [23–25] have attracted attention recently. Our main purpose here is to ask which one, if any, of the different theoretically possible descriptions of 1D anyons, is singled out as the dimensional reduction of the 2D theory.

The main candidates for real-world implementation of 2D anyon statistics remain the charge carriers of fractional quantum Hall systems [26–29] or their counterparts e.g. in cold atom emulations [30–35]. See [36–39] for reviews and [40, 41] for experimental evidence. Said charge carriers are 2D objects described in the bulk via the usual, aforementioned, anyon model [28, 29]. Much of fractional quantum Hall physics is however probed via the transport of charge carriers along 1D edge channels, which connects to our main question.

In another direction, the coupling of cold atoms to optical fields can lead, in the adiabatic limit, to the effective implementation of density-dependent gauge fields [30, 32, 42]. Key proposals in this direction have recently been experimentally realized [31, 43, 44]. In particular, signatures of the chiral BF model, connected to 1D Kundu anyons, have been observed [43, 45] by generating a magnetic-like *vector potential* proportional to matter density. On the other hand, the magnetic gauge picture of 2D anyons corresponds to a magnetic-like *field* proportional to matter density.

Indeed, in this note we explain (full mathematical details will be provided elsewhere) that the magnetic-gauge picture Hamiltonian for 2D anyons of statistical parameter $\alpha \in ]0, 2[$ converges, in the limit of a tight confinement along one spatial dimension, to the impenetrable boson model of the Tonks-Girardeau gas, soluble by Bose-Fermi mapping [46–48]. Thus, at leading order, the physics does not depend on $\alpha$ and is given by an extreme case of the Lieb-Liniger [49, 50] model. The behavior is always essentially fermionic [51].

These results might be interpreted in light of the enhanced effect of interactions in reduced dimensionalities. However, it is remarkable that the long-range magnetic interactions of the original model result in a limiting purely local theory. This finding is consistent with [52, 53] although our approach differs and seems more systematic. In particular it clarifies the vanishing of the long-range magnetic interaction. A particular, discontinuous, phase is acquired by the 2D wave-functions, gauging the interaction away when particles are aligned.

## 1 Models and main result

Consider a (multi-valued) wave function $\Psi : \left(\mathbb{R}^2\right)^N \to \mathbb{C}$ with anyonic exchange behavior, i.e.

$$\Psi(\mathbf{x}_1, \ldots, \mathbf{x}_j, \ldots, \mathbf{x}_k, \ldots, \mathbf{x}_N) = e^{i\alpha\pi}\Psi(\mathbf{x}_1, \ldots, \mathbf{x}_k, \ldots, \mathbf{x}_j, \ldots, \mathbf{x}_N), \tag{1}$$

with $\alpha \in ]0, 1]$ (by periodicity and complex conjugation, considering this range is sufficient). It is convenient to perform a singular gauge transformation

$$\Psi(\mathbf{x}_1, \ldots, \mathbf{x}_N) = \prod_{j<k} e^{i\alpha\phi_{jk}}\Phi(\mathbf{x}_1, \ldots, \mathbf{x}_N), \quad \text{with} \quad \phi_{jk} = \arg\frac{\mathbf{x}_j - \mathbf{x}_k}{|\mathbf{x}_j - \mathbf{x}_k|},$$

with $\Phi$ a bosonic wave function, symmetric under particle exchange. We have denoted $\arg(.)$ the angle of a planar vector with the horizontal axis. Applying this transformation, one finds

$$\left\langle \Psi \middle| \left(-i\nabla_{\mathbf{x}_j}\right)^2 \middle| \Psi \right\rangle = \left\langle \Phi \middle| D_{\mathbf{x}_j}^2 \middle| \Phi \right\rangle,$$

where the momentum operator for particle $j$ has changed as

$$-i\nabla_{\mathbf{x}_j} \rightsquigarrow D_{\mathbf{x}_j} := -i\nabla_{\mathbf{x}_j} + \alpha\mathbf{A}\left(\mathbf{x}_j\right), \tag{2}$$

with, denoting $(x, y)^\perp = (-y, x) \in \mathbb{R}^2$,

$$\mathbf{A}(\mathbf{x}_j) := \sum_{k \neq j} \frac{(\mathbf{x}_j - \mathbf{x}_k)^\perp}{|\mathbf{x}_j - \mathbf{x}_k|^2} \,. \tag{3}$$

In this picture we have traded the non-trivial exchange symmetry of wave-functions for a density-dependent magnetic field. Particle $j$ sees all the others as carrying an Aharonov-Bohm flux, leading to the magnetic field

$$B(\mathbf{x}_j) = \mathrm{curl}_{\mathbf{x}_j} \mathbf{A}(\mathbf{x}_j) = 2\pi\alpha \sum_{k \neq j} \delta_{\mathbf{x}_j = \mathbf{x}_k} \,.$$

We adopt this point of view throughout the note, using (the Friedrichs extension [54–59] of)

$$H_\varepsilon^{2\mathrm{D}} := \sum_{j=1}^N \left( D_{\mathbf{x}_j}^2 + V_\varepsilon(\mathbf{x}_j) \right) , \tag{4}$$

acting on bosonic wave-functions as our starting point, where

$$V_\varepsilon(\mathbf{x}) = V_\varepsilon(x, y) = x^2 + \frac{y^2}{\varepsilon^2} \,, \tag{5}$$

is a convenient way of enforcing 1D behavior along the horizontal axis in the limit $\varepsilon \to 0$. This is arguably a crude description (but, perhaps, also an instructive toy model) if one has a fractional quantum Hall edge in mind. There is however no difficulty in imposing such a potential on emerging anyons in cold atoms systems. The choice of a harmonic trapping is only out of convenience. Our results remain true with different choices, but the harmonic trapping has the virtue of leading to exactly soluble limit models.

We denote

$$\lambda_k^{2\mathrm{D}} = \min_{\substack{V_k \subset L^2(\mathbb{R}^{2N}) \\ \dim V_k = k}} \max_{\Psi \in V_k, \|\Psi\|_{L^2} = 1} \left\langle \Psi | H_\varepsilon^{2\mathrm{D}} | \Psi \right\rangle , \tag{6}$$

the eigenvalues of (4), defined by standard Courrant-Fisher min-max formulae. Let $\Psi_k$ be associated eigenfunctions, i.e.

$$H_\varepsilon^{2\mathrm{D}} \Psi_k = \lambda_k^{2\mathrm{D}} \Psi_k \,.$$

There can be degeneracies, in which case we count eigenvalues with their multiplicities.

For small $\varepsilon > 0$ one expects the motion in the two spatial directions to decouple (which is true only to some extent in this particular case, see below). The motion in the $y$ direction will be frozen in the ground state of the harmonic oscillator

$$H_\varepsilon^{\mathrm{HO}} := -\partial_y^2 + \frac{y^2}{\varepsilon^2} \,. \tag{7}$$

It turns out that the motion in the $x$ direction reduces to the free Hamiltonian

$$H^{1\mathrm{D}} := \sum_{j=1}^N -\partial_{x_j}^2 + x_j{}^2 \,, \tag{8}$$

but acting on the domain (of the Friedrichs extension)

$$\mathcal{D}^{1\mathrm{D}} := \left\{ \psi \in H^2(\mathbb{R}^N), \psi(x_j = x_k) \equiv 0, \text{ for all } j \neq k \right\} . \tag{9}$$

This restriction is equivalent to the addition of a delta pair-potential of infinite strength to (8). It is well-known that this impenetrable boson model can be mapped to a free fermionic one [46–

48]. In turn, this leads to an exact solution in the particular case above. However, our approach does not rely on this exact solution, and we could in fact have added extra interactions to our model. For simplicity, we do not consider this explicitly.

Let $e_\varepsilon$ and $u_\varepsilon$ be respectively the lowest eigenvalue and eigenfunction of (7). Let $(\lambda_k^{1D})_{k\in\mathbb{N}}$ be the eigenvalues of (8), with associated eigenfunctions $\psi_k, k = 1, 2\ldots$. In the model above, $\lambda_k^{1D}$ is a sum of eigenvalues of the harmonic oscillator $-\partial_x + x^2$ and

$$\psi_k = c_k \prod_{i<j} \mathrm{sgn}(x_i - x_j) \det_{i,j}(v_i(x_j)),$$

where $v_i$ are the associated one-particle eigenfunctions and $c_k$ is a normalization constant.

We state our main finding as a theorem.

**Theorem 1 (Dimensional reduction for anyons).**
*For all $k \in \mathbb{N}$, in the limit $\varepsilon \to 0$,*

$$\lambda_k^{2D} = N e_\varepsilon + \lambda_k^{1D} + o(1). \tag{10}$$

*Moreover, one can choose the 2D and 1D eigenbases $(\Psi_k)_k$ and $(\psi_k)_k$ in such a way that*

$$\int_{\mathbb{R}^{2N}} \left| \Psi_k - \psi_k(x_1, \ldots, x_N) \prod_{j=1}^N u_\varepsilon(y_j) \right|^2 \underset{\varepsilon\to 0}{\to} 0. \tag{11}$$

Although it seems from (11) that a standard decoupling between the two space directions takes place, the actual ansatz for the eigenfunctions $\Psi_k$ leading to the correct energy are more subtle. Essentially they are of the form

$$\psi_k(x_1, \ldots, x_N) \prod_{j=1}^N u_\varepsilon(y_j) \prod_{j<k} e^{-i\alpha S(\mathbf{x}_j - \mathbf{x}_k)}, \tag{12}$$

where we denote

$$S(\mathbf{x}) = S(x, y) = \arctan\left(\frac{y}{x}\right). \tag{13}$$

The above trial states have the correct bosonic symmetry because $S(\mathbf{x}) = S(-\mathbf{x})$, but they are not of the form "function of $x$ times function of $y$" that is more common in dimensional reductions. Note that $S(\mathbf{x})$ has a discontinuity along the line $x = 0$, so that it is crucial for (12) to be well-defined that $\psi_k(x_1, \ldots, x_N)$ vanishes whenever $x_j = x_k, j \neq k$.

The phase factors $e^{-i\alpha S(\mathbf{x}_j - \mathbf{x}_k)}$ modify the energy dramatically, gauging away the original magnetic interaction (see below). For this effect it is crucial to take advantage of the finite, albeit small, extension of our wave-guide, as shown by the discussion below.

## 2 A case for the Calogero-Sutherland model

Before we sketch the proof of the above, it is instructive to examine an argument that would rather point in the direction of the Calogero model with inverse square interactions (which is also a proposed model for 1D anyons) as effective description. This will emphasize two things:
• That quantization and dimensional reduction do not commute in this particular case. Classical particles with the above magnetic interactions would experience Calogero-like interactions if confined on a line.

• The role of the phase factors $e^{-i\alpha S(\mathbf{x}_j - \mathbf{x}_k)}$ in the main result. Indeed, if one chooses a simpler ansatz of the form

$$\psi_k = \phi(x_1, \dots, x_N) \prod_{j=1}^{N} u_\varepsilon(y_j),$$

the 1D function $\phi$ indeed experiences a Calogero-type Hamiltonian.

The possible connections between 2D anyons and Calogero-like models have already been pointed out in a similar context [60, 61]. It also arises for lowest Landau level anyons [52, 62–67] via very different mechanisms.

Consider $N$ classical particles with magnetic interactions akin to those of (4). We constrain them to move on the line $y = 0$, like pearls on a necklace. Expanding the square in (4), the Hamilton function for this system is

$$H(\mathbf{x}_1, \dots, \mathbf{x}_N; \mathbf{p}_1, \dots, \mathbf{p}_N) = \sum_{j=1}^{N} \left( |\mathbf{p}_j|^2 + x_j^2 \right)$$
$$+ 2\alpha \sum_{j \neq k} \mathbf{p}_j \cdot \frac{(\mathbf{x}_j - \mathbf{x}_k)^\perp}{|\mathbf{x}_j - \mathbf{x}_k|^2}$$
$$+ \alpha^2 \sum_{j \neq k \neq \ell} \frac{(\mathbf{x}_j - \mathbf{x}_k)^\perp}{|\mathbf{x}_j - \mathbf{x}_k|^2} \cdot \frac{(\mathbf{x}_j - \mathbf{x}_\ell)^\perp}{|\mathbf{x}_j - \mathbf{x}_\ell|^2}$$
$$+ \alpha^2 \sum_{j \neq k} \frac{1}{|\mathbf{x}_j - \mathbf{x}_k|^2},$$

where $\mathbf{p}_j = (p_j, 0)$ and $\mathbf{x}_j = (x_j, 0)$ are momenta and positions, respectively. The cross-term on the second line is clearly null. The term on the third line is null as well, as follows from grouping terms as in [68, Lemma 3.2]

$$\sum_{\text{cyclic in } 1,2,3} \frac{(\mathbf{x}_1 - \mathbf{x}_2)^\perp}{|\mathbf{x}_1 - \mathbf{x}_2|^2} \cdot \frac{(\mathbf{x}_1 - \mathbf{x}_3)^\perp}{|\mathbf{x}_1 - \mathbf{x}_3|^2} = \frac{1}{2\mathcal{R}(\mathbf{x}_1, \mathbf{x}_2, \mathbf{x}_3)^2},$$

with $\mathcal{R}(\mathbf{x}_1, \mathbf{x}_2, \mathbf{x}_3)$ the circumradius of the triangle with summits $\mathbf{x}_1, \mathbf{x}_2, \mathbf{x}_3$. This is the radius of the circle on which the three points lie, which is infinite for aligned points. Hence the Hamilton function boils down to

$$\sum_{j=1}^{N} \left( p_j^2 + x_j^2 \right) + \alpha^2 \sum_{j \neq k} \frac{1}{(x_j - x_k)^2},$$

which, once quantized, gives a Calogero Hamiltonian, albeit not with the expected $\alpha(\alpha - 1)$ coefficient [12, 13, 62] in front of the two-body term for particles of statistics parameter $\alpha$. Note that this reduction could in any case not be correct for all $\alpha$ because the 2D anyon energy is periodic in $\alpha$, but the Calogero energy is not [62, 69–71].

## 3   Argument for the main result

We turn to sketching the main insights of the proof of Theorem 1. Turning them into a rigorous mathematical proof is somewhat lengthy, and will be done elsewhere [72].

The crucial observation is that for particles close to the line $y = 0$, the vector potential of the Aharonov-Bohm fluxes in (4) can be gauged away. The vector potential

$$\mathbf{A}_0(\mathbf{x}) = \frac{\mathbf{x}^\perp}{|\mathbf{x}|^2} = \begin{pmatrix} -y \\ x \end{pmatrix} \frac{1}{x^2 + y^2},$$

for a unit Aharonov-Bohm flux at the origin has a non-zero curl and thus cannot be written as the gradient of a regular function globally. But, with $S$ defined as in (13)

$$\nabla S(\mathbf{x}) = \mathbf{A}_0(\mathbf{x}) - \begin{pmatrix} \pi\delta_{x=0}\mathrm{sgn}(y) \\ 0 \end{pmatrix}.$$

Hence, for any continuous function $\Psi(\mathbf{x})$ of finite kinetic energy vanishing on the line $x = 0$

$$\int_{\mathbb{R}^2} \left| (-i\nabla_{\mathbf{x}} + \alpha\mathbf{A}_0(\mathbf{x}))\left(\Psi(\mathbf{x})e^{-i\alpha S(\mathbf{x})}\right) \right|^2 d\mathbf{x} = \int_{\mathbb{R}^2} |\nabla\Psi|^2,$$

and this will be our model calculation (here performed in the relative coordinate of a particle pair).

The main point of our argument is the behavior

$$S(\mathbf{x}) \simeq_{|y|\ll|x|} \frac{y}{x}.$$

Indeed if one sets instead

$$\widetilde{S}(\mathbf{x}) := \frac{y}{x},$$

one finds

$$\nabla\widetilde{S}(\mathbf{x}) = \begin{pmatrix} -y/x^2 \\ 1/x \end{pmatrix} \simeq_{|y|\ll|x|} \mathbf{A}_0(\mathbf{x}),$$

and more precisely

$$\left|\nabla\widetilde{S} - \mathbf{A}_0\right| \le C\frac{|y|}{x^2}. \tag{14}$$

The singularity around $x = 0$ of the right-hand side would have to be tamed if we used $\widetilde{S}$ instead of $S$ in our trial state. Hence the latter choice is actually simpler, and we stick to it in the sequel.

Consider now a trial state $\Psi$ for (4) and write it as

$$\Psi(\mathbf{x}_1, \ldots, \mathbf{x}_N) = U_\varepsilon \prod_{j<k} e^{-\alpha i S(\mathbf{x}_j - \mathbf{x}_k)}\Phi, \tag{15}$$

$$U_\varepsilon(\mathbf{x}_1, \ldots, \mathbf{x}_N) = \prod_{j=1}^N u_\varepsilon(y_j),$$

with a new, continuous, unknown function $\Phi$ vanishing whenever $x_j = x_k, j \neq k$. A direct calculation yields

$$\left\langle\Psi|H_\varepsilon^{\mathrm{2D}}|\Psi\right\rangle = Ne_\varepsilon + \sum_j^N \int_{\mathbb{R}^{2N}} U_\varepsilon^2 x_j^2 |\Phi|^2 + \sum_j^N \int_{\mathbb{R}^{2N}} U_\varepsilon^2 \left|\nabla_{\mathbf{x}_j}\Phi\right|^2, \tag{16}$$

and we now seek critical points of this functional of $\Phi$. For energy upper bounds it will clearly be favorable for $\Phi$ not to depend on $y_1, \ldots, y_N$ and we then recognize the energy functional corresponding to (8).

We then need to prove a lower bound of the correct form for the energy of a true eigenstate $\Psi$ of the 2D model. We do not know a priori that the corresponding $\Phi$ vanishes when $x_j = x_k, j \neq k$, but (16) can in this case be replaced by

$$\left\langle\Psi|H_\varepsilon^{\mathrm{2D}}|\Psi\right\rangle \ge Ne_\varepsilon + \sum_j^N \int_{\Lambda_\eta} U_\varepsilon^2 x_j^2 |\Phi|^2 + \sum_j^N \int_{\Lambda_\eta} U_\varepsilon^2 \left|\nabla_{\mathbf{x}_j}\Phi\right|^2, \tag{17}$$

where

$$\Lambda_\eta = \left\{ (\mathbf{x}_1, \ldots, \mathbf{x}_N) \in \mathbb{R}^{2N}, \ |x_j - x_k| > \eta, \ \forall j \neq k \right\}.$$

Extracting the singular phase is unproblematic on the latter set, and we can then pass to the limit in the above, $\varepsilon \to 0$ and $\eta \to 0$, obtaining an energy lower bound essentially of the desired form. More precisely, in [72, Proposition 4.3] we prove the following:

**Lemma 2.** *Let*

$$\phi(x_1, y_1, \ldots, x_N, y_N) := \Phi(x_1, \sqrt{\varepsilon} y_1, \ldots, x_N, \sqrt{\varepsilon} y_N). \tag{18}$$

*After possibly extracting a subsequence,*

$$\phi \underset{\varepsilon \to 0}{\to} \psi_0,$$

*where the limit $\psi_0$ has no dependence on the y-coordinates, and satisfies*

$$\liminf_{\varepsilon \to 0} \left( \langle \Psi | H_\varepsilon^{2\mathrm{D}} | \Psi \rangle - N e_\varepsilon \right) \geq \sum_{j=1}^N \left( \int_{\mathbb{R}^N} \left| \partial_{x_j} \psi_0 \right|^2 + \int_{\mathbb{R}^N} |x_j|^2 |\psi_0|^2 \right). \tag{19}$$

The main difficulty is now to ensure some form of vanishing around particle encounters for our limit model to indeed be set on (9), i.e. that the 1D function obtained by passing to the limit has finite kinetic energy over the whole space (including across diagonals). To this end we use the following Hardy-like inequality: For any $\Psi$, with the modified momentum as in (2)

$$\sum_{j=1}^N \left\langle \Psi | \left( D_{\mathbf{x}_j} \right)^2 | \Psi \right\rangle \geq c_{\alpha,N} \int_{\mathbb{R}^{2N}} \sum_{j<k} \frac{1}{|\mathbf{x}_j - \mathbf{x}_k|^2} |\Psi(\mathbf{x}_1, \ldots, \mathbf{x}_N)|^2, \tag{20}$$

where the best possible constant $c_{\alpha,N}$ depends only on $\alpha$ and $N$. Such an inequality originates from [55, 56, 68, 73] (see also [74] for review and generalizations) where it is proved in particular that

- $c_{\alpha,N} \geq c N^{-1}$ with a universal $c > 0$ if $\alpha$ is an odd-numerator fraction.

- $c_{\alpha,2} > 0$ for any $\alpha \neq 0$.

We improve these bounds by proving that there exists a universal constant $c' > 0$ such that

$$c_{\alpha,N} \geq c' N^{-2}, \quad \text{for any} \quad \alpha \neq 0,$$

which leads to our main result by providing the desired vanishing in the whole parameter range. One can understand heuristically why by considering the contribution of the set where $|x_j - x_k| \lesssim \sqrt{\varepsilon}$ and $|y_\ell| \lesssim \sqrt{\varepsilon}$ for $\ell = 1 \ldots N$ to the right-hand side of (20). One has $|\mathbf{x}_j - \mathbf{x}_k|^2 \lesssim \varepsilon$ on this set. If the limiting 1D function does not vanish for $x_j \sim x_k$, then $|\Psi|^2 \propto U_\varepsilon^2 \propto \varepsilon^{-N/2}$. The total contribution (volume times typical value of the integrand) would thus be of order $\varepsilon^{-1/2}$, much larger than the expected energies, of order unity after removal of the contribution of $N e_\varepsilon$ as in (10). More precisely this leads us to (see Lemma 4.8 in [72])

**Lemma 3.** *Let*

$$\psi(x_1, \ldots, x_N) := \int_{\mathbb{R}^N} \mathrm{d}y_1 \cdots \mathrm{d}y_N \, |\phi(\mathbf{x}_1, \ldots, \mathbf{x}_N)| \, U_1^2(y_1, \ldots, y_N) \mathbb{1}_{|y_1| \leq 1/4} \cdots \mathbb{1}_{|y_N| \leq 1/4}, \tag{21}$$

*and $K$ be a bounded open subset in $\mathbb{R}^{N-1}$. For a constant $\gamma \geq 0$, we define*

$$K_\gamma := \left\{ (x_2, x_3, \ldots, x_N) \in K : |x_2 - x_j| > \sqrt{\gamma}/2, \ \forall j \geq 3 \right\}.$$

*Then when $\gamma \geq \varepsilon$, we have*

$$\int_{K_\gamma} |\psi(x_2, x_2, x_3, \ldots, x_N)|^2 \, \mathrm{d}x_2 \cdots \mathrm{d}x_N \leq C_K \varepsilon^{\frac{s}{2}},$$

*for some constants $C_K > 0$ and $s \in (0, 1)$ both independent of $\varepsilon$.*

The above will, after passing to the limit, guarantee that $\psi_0$ vanishes when two arguments come close, the others being at least at a distance $\sqrt{\gamma}/2$. One can finally pass to the limit $\gamma \to 0$ to deduce that $\psi_0(x_j = x_k) \equiv 0$ for any $k \neq j$. Hence indeed the energy bounds force the limiting 1D function to vanish upon particle encounters, as desired.

# 4  Conclusions

We have studied 2D anyons of statistics parameter $\alpha$ in the magnetic gauge picture, i.e. seeing them as bosons in a varying magnetic field proportional to matter density. In a cold atoms context this corresponds to proposals made e.g. in [32]. We imposed a dimensional reduction by ways of a strongly anisotropic trap, as in recent cold-atoms experiments probing density-dependent gauge theories [31, 43, 44].

In the 1D limit we found that a suitable choice of gauge removes long-range magnetic interactions. Their only remnant is a hard-core condition upon particle encounters, leading to the Girardeau-Tonks model of 1D bosons for any $\alpha \neq 0$. Non-trivial dependence on $\alpha$ might survive at sub-leading order, in which case it could be determined by perturbation theory around Girardeau's solution of the impenetrable 1D Bose gas.

# Acknowledgments

We had interesting discussions with Douglas Lundholm, Michele Correggi, Per Moosavi and Nikolaj Thomas Zinner.

**Funding information**   Funding from the European Research Council (ERC) under the European Union's Horizon 2020 Research and Innovation Programme (Grant agreement COR-FRONMAT No 758620) is gratefully acknowledged.

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
