# Peer review of "Anyons in a tight wave-guide and the Tonks-Girardeau gas"

_SciPost Physics Core, doi:SciPost Phys. Core 6, 079 (2023)_

## Round 1 · Referee Report · Anonymous (Referee 1) · 2023-6-6

Report

The authors consider the problem of dimensional reduction for a model of 2D non-interacting anyons. Namely, starting from a 2D system, they consider adding a strong confining potential along one dimension to constrain the motion along the other. Their main result is a mathematical theorem. It states that, in the limit of very strong confining potential, the dynamics is mapped onto a collection of one dimensional systems described by the Tonks-Girardeau Hamiltonian. In particular, there is no dependence on the original exchange statistics parameter, so that the one-dimensional systems are always trivially fermionic.

I believe the draft is well written and clear. However, I am a bit hesitant to recommend publication for the reasons below:

1) First, the main physical message was already found, or at least strongly suggested by, other works in the literature almost 30 years ago, as also mentioned by the authors (see Ref. [31], [63]). Therefore, the main contribution of the work is mainly technical, consisting of the mathematical proof of the theorem.

2) On the other hand, in this letter-looking draft, all the mathematical details are omitted. I believe this choice would be justified in the case the result by itself is new, or when the details of the calculations hide the main physical discussions. Here, however, it seems to me the mathematical theorem is the main result. Therefore, given the scope of Scipost, I would find it hard to recommend publication of the draft as is, without substantiating the technical contributions. This could be done by adding one or more appendices.

3) Finally, in my opinion the topic is not so timely. The authors have described one experimental protocol to reduce a 2D anyonic system into 1D ones. As I said, similar conclusions were put forward already in the 90's (see Ref. [63]). However, in the past decades different experimental proposals have been put forward to realize genuine anyonic statistic in one-dimensional cold-atomic systems (see for instance the very recent paper https://arxiv.org/abs/2306.01737). I think it would be useful if the authors could discuss a bit more how this work relates to these recent developments.

In any case, I don't think the paper is suitable for Scipost Physics, and I believe Scipost Physics Core would be a more appropriate choice for this submission.

  • validity: -
  • significance: -
  • originality: -
  • clarity: -
  • formatting: -
  • grammar: -

Author:  Nicolas Rougerie  on 2023-06-16  [id 3738]

(in reply to Report 1 on 2023-06-06)

We thank very much the referee for his/her report. We would like to argue that our paper is suitable for publication in SciPost Physics, despite the constructive criticism that was formulated. We briefly answer to the latter below.

" 1) First, the main physical message was already found, or at least strongly suggested by, other works in the literature almost 30 years ago, as also mentioned by the authors (see Ref. [31], [63]). Therefore, the main contribution of the work is mainly technical, consisting of the mathematical proof of the theorem. "

Upon exchanging about the topic at hand with many experts of the field, we feel that the mentioned older references did not fully convince the community, even at the level of mathematical rigor typically expected in theoretical physics. The papers are not very well-known (one of them exists only as a preprint), and the arguments do not seem to have percolated. This could be due to the fact that these papers more or less directly assume that the limit model must be Lieb-Liniger like, and then proceed to compute the coupling constant, which they argue must diverge, a point on which we agree.

We think that our main contribution (besides the rigorous mathematical proof) is to explain, without ad-hoc assumptions, the mechanism turning the non-local 2D anyons model to a fully local 1D Lieb-Liniger-like one. This is achieved by the identification of

  1. the correct gauge transformation effectively removing the self-consistent magnetic field, which is well-defined if 1D trial states are fermionic.

  2. the fact that an improved Hardy inequality guarantees the desired fermionic vanishing upon particle encounter.

Our impression is that the previous references, falling short of emphasizing these mechanisms, are not fully convincing. Perhaps that was not emphasized clearly enough in our text, but we do not think that such remarks, which might be interpreted as criticism, have their place there.

" 2) On the other hand, in this letter-looking draft, all the mathematical details are omitted. I believe this choice would be justified in the case the result by itself is new, or when the details of the calculations hide the main physical discussions. Here, however, it seems to me the mathematical theorem is the main result. Therefore, given the scope of Scipost, I would find it hard to recommend publication of the draft as is, without substantiating the technical contributions. This could be done by adding one or more appendices. "

We could probably squeeze the proofs in an appendix, but that would be at the expense of readability, and thus we would still present a longer paper, such as our https://arxiv.org/abs/2305.06670, which is 32 pages long.

We also think that some of the details of the proof do indeed obscure the main physical discussion. For example, the heuristics presented in the last paragraph before Section 4 are (in our opinion) rather convincing at the level of rigor the main physics literature aims at. Backing then with mathematically rigorous estimates takes several pages, and requires arguments that are not typically well-known in the condensed matter physics community, cf https://arxiv.org/abs/2305.06670 again.

" 3) Finally, in my opinion the topic is not so timely. The authors have described one experimental protocol to reduce a 2D anyonic system into 1D ones. As I said, similar conclusions were put forward already in the 90's (see Ref. [63]). However, in the past decades different experimental proposals have been put forward to realize genuine anyonic statistic in one-dimensional cold-atomic systems (see for instance the very recent paper https://arxiv.org/abs/2306.01737). I think it would be useful if the authors could discuss a bit more how this work relates to these recent developments. "

In the next version of our paper we will certainly, discuss the experimental work https://arxiv.org/abs/2306.01737, which was posted after our preprint. In the current version we already comment on the connection with Refs 12 and 24. We think this discussion is timely indeed, because these experiments observe striking properties of 1D anyons, but claim that the model they implement corresponds to the 1D reduction of the 2D anyon model, which we prove is not the case. This does not, in our opinion, diminish the importance of Refs 12 and 24 (1D and 2D anyons are just not so directly connected) but shows two things

  1. an investigation of the 2D to 1D limit is tricky, so that a rigorous theorem is desirable to arbitrate between different claims.

  2. the fact that 2D anyons fermionize in 1D is not widely known, so that a thorough discussion is useful at the very time where 1D anyon physics becomes accessible to experiments.

We do not overly emphasize these points in the text, in order for it not to be mistaken for a criticism of Refs 12 and 24, which it is not.

A last point is that the proposed protocol of Ref 67, if coupled with a 1D confinement, could lead to the observation of the phenomena we discuss theoretically. We ignore wether experimental groups are currently working in this direction, so we stayed cautions in this regard.

CONCLUSION:

We feel that our paper fully vindicates a crucial mechanism in anyon physics. It is true that it was already hinted at in the literature, but not in the precise form we discuss. We emphasize that this concerns not only the mathematical proof, but also its' basic physical ingredients. This discussion seems timely to us, in view of the developments on the experimental side, one of which occured just at the time where we were preparing the manuscript, and another a few months after we posted it.

We remain at the disposal of the editors and referees if they feel further discussion is required, or if a new version could be resubmitted to SciPost physics on the grounds of the above elements.

---

## Round 1 · Referee Report · Anonymous (Referee 2) · 2023-6-17

Report

This manuscript explores the physics of 1D systems having fractional statistics which interpolates between usual bosonic and fermionic ones on varying of a statistical parameter α. As the authors pointed out in the introduction, there have been several proposals to obtain anyonic models in one dimension. Several studies focused on e.g. 1D bosonic systems subject to a density-dependent hopping in a way that the resulting 1D model is effectively described by anyonic statistics. On the other hand, in two spatial dimensions, anyonic statistics can naturally emerge from the braiding of two particles, or equivalently, from the Aharonov-Bohm phase picked by the wavefunction in the presence of magnetic flux tubes.

In this context, the authors wish to clarify whether the proposals for 1D anyonic systems can be thought of as a dimensional reduction of 2D anyonic models. In particular, they consider a 2D anyonic model and impose a dimensional reduction by means of a strongly anisotropic trapping potential to obtain an effective 1D model. Their main result is that, independently of α, the resulting 1D model is described by strongly repulsive bosons (Tonks-Girardeau gas).

I have found the paper well-written and scientifically sound, although I haven’t gone into the details of the authors’ proof. I believe that this article deserves publication in some form but I have some perplexities regarding its publications in SciPost Physics. In my opinion, this manuscript would better fit a more specialized journal on mathematical physics.

Indeed, I think that the majority of recent (and less recent) studies on 1D anyons are focused on the consequences of having fractional statistics in one dimension (whose physics quite differs from 2D) rather than relating 1D to 2D anyonic models. Moreover, several studies proposed techniques to experimentally engineer 1D anyons (e.g. Cardarelli et al., Phys. Rev. A 94, 023615 (2016); Keilmann et al., Nat. Commun. 2, 361 (2011) ).

To conclude, I believe that the manuscript would benefit if the authors decide to include more mathematical details on the proof for a more specialized audience. In its current form, I think that the manuscript does not meet the acceptance criteria required for publishing in SciPost Physics.

---

## Round 2 · Author Response

Dear editor
We have modified the text in the direction of more mathematical details, for a slightly longer text in SciPost Physics Core.
Sorry that we did not find the time to deal with this until now.
with best regards
NR and QY
We have modified the text in the direction of more mathematical details, for a slightly longer text in SciPost Physics Core.
Sorry that we did not find the time to deal with this until now.
with best regards
NR and QY

---

## Round 2 · List of Changes

The main changes are the statements of Lemma 2 and Lemma 3, with the associated discussion.
References have also been updated.
References have also been updated.

---

## Editorial Decision

published